# Improved SARS-CoV-2 Neutralization of Delta and Omicron BA.1 Variants of Concern after Fourth Vaccination in Hemodialysis Patients

**DOI:** 10.3390/vaccines10081328

**Published:** 2022-08-16

**Authors:** Cho-Chin Cheng, Louise Platen, Catharina Christa, Myriam Tellenbach, Verena Kappler, Romina Bester, Bo-Hung Liao, Christopher Holzmann-Littig, Maia Werz, Emely Schönhals, Eva Platen, Peter Eggerer, Laëtitia Tréguer, Claudius Küchle, Christoph Schmaderer, Uwe Heemann, Lutz Renders, Ulrike Protzer, Matthias Christoph Braunisch

**Affiliations:** 1Institute of Virology, School of Medicine, Technical University of Munich, 81675 Munich, Germany; 2Department of Nephrology, School of Medicine, Technical University of Munich, Klinikum Rechts der Isar, 81675 Munich, Germany; 3TUM Medical Education Center, School of Medicine, Technical University of Munich, 81675 Munich, Germany; 4Kidney Center Eifel Dialyse, 53894 Mechernich, Germany; 5KfH Kidney Center Harlaching, Munich-Harlaching, 81545 Munich, Germany; 6KfH Kidney Center, 83278 Traunstein, Germany; 7German Center for Infection Research (DZIF), Partner Site, 81675 Munich, Germany; 8Institute of Virology, Helmholtz Munich, 85764 Munich, Germany

**Keywords:** hemodialysis, SARS-CoV-2, COVID-19 vaccination, in-vitro viral neutralization

## Abstract

Hemodialysis patients are exposed to a markedly increased risk when infected with SARS-CoV-2. To date, it is unclear if hemodialysis patients benefit from four vaccinations. A total of 142 hemodialysis patients received four COVID-19 vaccinations until March 2022. RDB binding antibody titers were determined in a competitive surrogate neutralization assay. Vero-E6 cells were infected with SARS-CoV-2 variants of concern (VoC), Delta (B.1.617.2), or Omicron (B.1.1.529, sub-lineage BA.1) to determine serum infection neutralization capacity. Four weeks after the fourth vaccination, serum infection neutralization capacity significantly increased from a 50% inhibitory concentration (IC50, serum dilution factor 1:x) of 247.0 (46.3–1560.8) to 2560.0 (1174.0–2560.0) for the Delta VoC, and from 37.5 (20.0–198.8) to 668.5 (182.2–2560.0) for the Omicron VoC (each *p* < 0.001) compared to four months after the third vaccination. A significant increase in the neutralization capacity was even observed for patients with high antibody titers after three vaccinations (*p* < 0.001). Ten patients with SARS-CoV-2 breakthrough infection after the first blood sampling had by trend lower prior neutralization capacity for Omicron (*p* = 0.051). Our findings suggest that hemodialysis patients benefit from a fourth vaccination in particular in the light of the highly infectious SARS-CoV-2 Omicron-variants. A routinely applied four-time vaccination seems to broaden immunity against variants and would be recommended in hemodialysis patients.

## 1. Introduction

In hemodialysis patients, a SARS-CoV-2 infection is associated with a markedly increased morbidity and mortality compared to the general population, with a mortality rate of more than 20% in hospitalized patients [1,2,3]. In the last two years, we have learned that double vaccination might not be enough to achieve adequate long-term immune protection in all hemodialysis patients, and triple vaccination offers significantly better protection against COVID-19 in this patient group [1,4]. Even if an infection cannot always be prevented, the course of the COVID-19 disease, in general, is milder depending on the number of vaccinations in hemodialysis patients [5]. However, even in hemodialysis patients with an inadequate immune response after multiple vaccinations, morbidity remains significantly increased [6].

A third vaccination is associated with an increased virus neutralization capacity in the general population [7,8]. Therefore, to date, the third vaccination has become part of the standard vaccination regimen, and meanwhile, a fourth vaccination is recommended in risk groups like hemodialysis patients [9]. The usefulness of a third and now a fourth vaccination is based on data from the general population and was obtained during the SARS-CoV-2 Delta wave. However, infections with the Omicron variant of concern (VoC) dramatically increased in 2022 [10]. Therefore, the question remains whether the currently recommended vaccination regimen in hemodialysis patients also offers effective protection towards VoC Omicron.

The aim of this study was to investigate whether hemodialysis patients benefit from a fourth vaccination and if the immune response after the fourth vaccination has a comparable efficacy towards the VoCs Delta and Omicron BA.1.

Here, we present the results of the live-virus infection neutralization of SARS-CoV-2 Delta and Omicron BA.1 VoCs and antibody-mediated immunity shortly before compared to four weeks after the fourth COVID-19 vaccination in a cohort of 142 hemodialysis patients.

## 2. Materials and Methods

### 2.1. Study Design

The COVIIMP study (German: “COVID-19-Impfansprechen immunsupprimierter Patient*innen”) is a prospective observational study examining the COVID-19 immunization success and the clinical course of COVID-19 in patients immunocompromised due to kidney transplantation, a rheumatologic disease, or dialysis who received immunization against SARS-CoV-2 as recommended by the German health authorities.

All participants provided written informed consent. The study, conforming to the ethical guidelines of the Helsinki Declaration, was approved by the Medical Ethics Committee of the Klinikum rechts der Isar of the Technical University of Munich (approval number 163/21 S-SR, 19 March 2021) and registered at the Paul Ehrlich Institute (NIS592).

### 2.2. Study Population

Of 513 enrolled patients, 142 patients requiring maintenance hemodialysis were selected. These patients received four COVID-19 vaccinations between 19 December 2020 and 20 March 2022 and underwent blood analysis before and after the fourth vaccination (Figure 1A). This subpopulation was recruited in four dialysis centers (Klinikum rechts der Isar, KfH Kidney Center Traunstein, Kidney Center Eifeldialyse, KfH Kidney Center München-Harlaching). Demographic data, medical history, history of transplantation, and comorbidities as assessed by the Charlson Comorbidity Index (CCI) were collected. Immunosuppressive medication during the vaccination period was documented.

### 2.3. Hepatitis B Vaccination

Hepatitis B vaccination status was based on medical reports and, if available, serological laboratory data on anti-HBs antibodies. Patients were considered non-responders if an anti- HB titer below 10 IU/l despite three hepatitis B vaccinations was documented or their treating physicians classified them as a hepatitis B non-responder, according to local standards.

### 2.4. SARS-CoV-2 Infection

We identified participants as SARS-CoV-2 convalescent if they had a prior positive SARS-CoV-2 PCR or at least one positive serological SARS-CoV-2 nucleocapsid-specific IgG measurement [4,11].

### 2.5. Sample Collection

Blood was collected for analysis in a median two (2.0–3.25) days before (analysis 1) and 26 (26.0–26.0) days after (analysis 2) the fourth vaccination.

### 2.6. SARS-CoV-2 IgG Assay

Antibodies in patients’ sera were detected using commercial surrogate paramagnetic particle chemiluminescence immunoassays (CLIA, Yhlo Biotechnology, Shenzhen, China) performed on the iFlash 1800 platform. Nucleocapsid-specific IgG antibodies (anti-N IgG) were determined using the 2019-nCoV IgG kit. The surrogate neutralization assay (NAb) was performed with the iFlash 2019-nCoV NAb kit based on the competition of serum antibodies with recombinant angiotensin-converting enzyme 2 for binding the SARS-CoV-2 Wuhan strain receptor binding domain (RBD) and has been adapted for quantification to manufacturer’s instructions [11,12]. The cut-off level for seropositivity was set at 10 neutralizing units per milliliter (AU/ml) according to the manufacturer’s instructions. Surrogate neutralization activity expressed as AU/ml can be adapted to WHO standard (AU/mL × 2.4 = BAU/mL [binding units/mL]). The maximum measurable value for NAb was 800 AU/mL, lower level of detection was 4 AU/mL. If values exceeded the upper limit of quantification, a value of 801 AU/mL was used for statistical analysis. NAb high-response was defined as levels ≥700 AU/mL before the fourth vaccination. N-specific IgGs ≥ 10 AU/mL were qualitatively determined as reactive. In one patient analysis of NAbs after the fourth vaccination was missing.

### 2.7. SARS-CoV-2 Infection-Neutralization Assay

Serum infection-neutralization capacity was analyzed as previously described [8]. Briefly, SARS-CoV-2 isolates were isolated from nasopharyngeal swabs of COVID-19 infected individuals. To obtain a high titer of virus stock, Vero-E6 cells were infected with VoC Delta (B.1.617.2, GISAID EPI ISL: 2772700) or Omicron (B.1.1.529, sub-lineage BA.1, GISAID EPI ISL: 7808190) and incubated in Dulbecco’s modified Eagle’s medium. After 2–3 days following inoculation, the cell culture medium was collected, centrifuged, and the virus-containing supernatant was stored at −80 °C. Prior to the neutralization experiments, viral titers were verified by plaque assay, and strain identity was confirmed by next-generation sequencing. All measurements were performed using serum samples stored at −80 °C and defrosted and stored at 4 °C on the day before the analysis. Samples from all patients were analyzed in parallel. For quantification of the neutralization capacity, two-fold serial dilutions of the sera from 1:20 to 1:2560 were incubated with a predefined multiplicity of infection (MOI) of 0.03 (450 PFU/15,000 cells/well) of either of the VoCs for 1 h at 37 °C. The MOI was determined from an in-cell ELISA pre-test by which we observed viral signal saturation 24 h after infection. After the 1-h inoculation, the inoculum was transferred onto pre-seeded Vero E6 cells for another one-hour incubation at 37 °C. The infection was terminated after one day and followed by an in-cell ELISA to detect SARS-CoV-2 N-protein. Cells were fixed with 4% paraformaldehyde and permeabilized by 0.5% saponin buffer. After blocking with 10% goat serum, cells were stained using anti-SARS-CoV-2-N primary (40143-T62, Sino Biological, Beijing, China) and a goat anti-rabbit IgG2a-HRP secondary antibody (EMD Millipore/#12-348, Shanghai, China), and eventually transformed into a colorimetric signal by adding substrate tetramethylbenzidine (TMB). To determine serum IC50 values, a nonlinear regression curve was applied, and the dilution factor at which 50% inhibition was observed and calculated using PRISM software (GraphPad, Shanghai, China). Patients were classified as low or non-responders if the IC50 value of the infection neutralization was ≤1:20.

### 2.8. Statistical Analysis

Categorical variables are presented as frequencies and percentages. Continuous variables are expressed as mean ± standard deviation (SD) or median and interquartile range (IQR), as appropriate. Group differences were tested with the χ^2^ test or Fisher test. The independent samples *t*-test or Mann-Whitney-U test was used for continuous variables, as appropriate. Paired samples were examined with the Wilcoxon test and the McNemar test, as appropriate. Spearman correlation was used for correlation analysis.

Univariate and multivariate linear regression models were applied to identify possible predictors of the infection-neutralizing capacity of VoC Delta or Omicron BA.1 (IC50) out of the following candidate variables: age, dialysis vintage, presence of immunosuppression, comorbidities, and hepatitis B vaccination non-response. Possible predictors were preselected prior to the statistical analysis. Logistic regression was used to examine the neutralization capacity towards an infection with SARS-CoV-2.

All tests were conducted two-sided, and *p* < 0.05 was considered significant. Statistical analysis was performed using R version 4.0.2 (R Foundation for Statistical Computing, Vienna, Austria).

## 3. Results

Overall, 142 patients on maintenance hemodialysis were included (Figure 1A). Patients had a median age of 72.6 (61.5–80.6) years. 48 (33.8%) patients were female. The median dialysis vintage was 48.9 (21.3–83.7) months. At the time of the first, second, third, and fourth vaccination, 124 (87.3%), 125 (88.0%), 136 (95.8%), and 142 (100%) were on maintenance hemodialysis, respectively. Further details of patient characteristics can be found in Table 1 for all patients and stratified by infection neutralization response against VoC Omicron BA.1.

### 3.1. COVID-19 and Vaccinations

All patients received four vaccinations, eight and six of which received their first and second vaccination with AZD1222 (Vaxzevria^®^, AstraZeneca Canada Inc., Mississauga, ON, Canada) by AstraZeneca. All other vaccinations were done with mRNA-based vaccines (BNT162b2 by BioNTech-Pfizer, New York, NY, USA or mRNA-1273 by Moderna, Cambridge, MA, USA). Fifteen patients received two or more vaccinations with mRNA-1273 (Spikevax^®^, Moderna), and the remaining patients received BNT162b2 (Comirnaty^®^, BioNTech-Pfizer). The median duration between the first and the fourth vaccination was 338.0 (333.0–342.0) days, and between the third and the fourth vaccination 126.0 (105.0–126.0) days, respectively. The median duration between the third vaccination and the first blood sampling was 4.1 (3.4–4.1) months.

A SARS-CoV-2 breakthrough infection indicated by SARS-CoV-2 nucleocapsid-specific IgG antibody positivity or a positive SARS-CoV-2 PCR occurred in 22 (15.5%) individuals before the second blood sampling after the fourth vaccination (Figure 1B). In these patients, the average time between the SARS-CoV-2 infection and the second blood collection was 215.7 ± 223.3 days. Of these, seven patients had no known history of SARS-CoV-2 infection but were classified as convalescent due to positive anti-nucleocapsid IgG detection. Four (18.2%) of the 22 infected patients were treated with SARS-CoV-2-specific monoclonal antibodies. Ten (7.0%) patients had a SARS-CoV-2 infection between the two blood drawings before and after the fourth vaccination. No patient reported recurrent SARS-CoV-2 infections (Figure 1B).

### 3.2. Immunosuppression

Immunosuppressive medication was prescribed in 16 (11.3%) patients during the observation period. Reasons for immunosuppression were history of organ transplantation in eight, cancer treatment in four, underlying kidney disease in two, and unknown causes in two other patients. Immunosuppressive agents were glucocorticoids in 14, tacrolimus in four, mycophenolate mofetil in three, and others in two patients (lenalidomide, rituximab, and reduced dose CHOP).

### 3.3. Impact of Four Vaccinations on Neutralization Capacity and NAbs

After the fourth vaccination significantly increased serum neutralization capacities were found for both VoCs, Delta and Omicron BA.1. Infection neutralization capacity for Delta increased from a median IC50 (serum dilution factor, 1:x) of 247.0 (46.3–1560.8) to 2560.0 (1174.0–2560.0), and for Omicron BA.1 from 37.5 (20.0–198.8) to 668.5 (182.2–2560.0) (each *p* < 0.001) (Figure 2A,B). NAb levels significantly increased from 721.0 (184.5–801.0) to 801.0 (801.0–801.0, *p* < 0.001) (Figure 2C). Serum neutralization capacity after the fourth vaccination was significantly lower for Omicron BA.1 compared to Delta (668.5 [182.2–2560.0] vs. 2560.0 [1174.0–2560.0], *p* < 0.001). Similar to the overall cohort, when analyzing only NAb high-responder, we found a significant increase for the neutralization capacity for both VoCs, Delta (1172.5 [382.8–2560.0] vs. 2560.0 [2560.0–2560.0], *p* < 0.001) and Omicron BA.1 (170.5 [56.3–468.5] vs. 2553.0 [640.2–2560.0], *p* < 0.001).

Patients with a serum IC50 ≤ 20 were classified as low, and those with no detectable neutralization as non-responder. Regarding Delta and Omicron BA.1 infection neutralization capacity, significantly fewer patients (Delta: 30 vs. 5; Omicron BA.1: 61 vs. 12, each *p* < 0.001) were low or non-responders after the fourth vaccination. The percentage of NAb responders was already very high before the fourth vaccination and did not further increase significantly (136 [95.8%] vs. 139 [98.6%], *p* = 0.13) (Figure 3).

After the fourth vaccination, infection neutralization of Delta and NAb titers were correlated highly significantly (*p* < 0.0001) but moderately (rho = 0.50) positive. Similarly, the correlation of the infection neutralization capacity of Omicron BA.1 and NAb was highly significant (*p* < 0.0001) and moderately (rho = 0.44) positive.

Univariate regression analysis showed significantly reduced neutralization capacity for Delta after the fourth vaccination if immunosuppressive medication (*p* = 0.001) or hepatitis B vaccination non-response (*p* = 0.046) was present (Table 2A, left column). Multivariate analysis showed a reduced Delta neutralization capacity after the fourth vaccination if immunosuppressive medication (*p* < 0.001) was taken and–by trend–if hepatitis B vaccination non-response was present (*p* = 0.070) (Table 2A, right column). For Omicron BA.1 infection neutralization, no such association was present (Table 2B). Univariate and multivariate analyses showed reduced NAbs after the fourth vaccination if immunosuppressive medication was prescribed (Table 2C).

When comparing serum neutralizing capacities after the fourth vaccination between subgroups we saw significant differences in Delta infection neutralization if immunosuppression was prescribed (716.5 [176.2–2560.0] vs. 2560.0 [1678.0–2560.0], *p* = 0.002) (Figure 4A), and by trend for Omicron BA.1 (193.5 [80.0–1481.8] vs. 820.5 [214.3–2560.0], *p* = 0.067) (Figure 4B). Patients with a history of SARS-CoV-2 infection had by trend a higher IC50 value for Delta (2560.0 [2560.0–2560.0] vs. 2560.0 [955.2–2560.0], *p* = 0.069) and significantly higher values for Omicron BA.1 neutralization (1952.0 [893.2–2560.0] vs. 489.0 [157.8–2560.0], *p* = 0.013) (Figure 4C,D). If patients were classified as hepatitis B vaccine non-responder, they had significantly lower IC50 values for Delta neutralization (2460 [531.0–2560.0] vs. 2560.0 [1765.0–2560.0], *p* = 0.018) (Figure 4E), but not for Omicron BA.1 neutralization (553.0 [103.5–1762.5] vs. 760 [254.0–2560.0], *p* = 0.18) (Figure 4F).

### 3.4. Impact of NAb and Infection Neutralization Capacity on Breakthrough Infections

Finally, the ten patients with a SARS-CoV-2 breakthrough infection between the first and the second blood sampling had by trend lower serum neutralization capacity for Omicron BA.1 at the first blood sampling being almost significant (10.0 [0.0–26.8] vs. 42.5 [20.0–217.5], *p* = 0.051) (Figure 5). No difference was detected for serum neutralization capacity of Delta (189.5 [42.5–1167.0] vs. 257.5 [50.8–1583.8], *p* = 0.54). The VoC causing the SARS-CoV-2 infection was not determined. Omicron BA.1 serum neutralization capacity at the first blood sampling could not predict the COVID-19 breakthrough infection between the first and the second blood sampling (*p* = 0.29) when using univariate logistic regression.

## 4. Discussion

This prospective observational study demonstrates that hemodialysis patients benefit from a fourth COVID-19 vaccination. Serum infection neutralization capacity increased more than 10-fold for Delta and almost 18-fold for Omicron BA.1 after a fourth vaccination indicating better protection from infection with these highly infectious SARS-CoV-2 VoCs. The strength of our study is the examination of the live-virus infection neutralization capacity of patients’ sera for two of the most recent SARS-CoV-2 VoCs, Delta and Omicron BA.1. These two variants are also most distant from the original SARS-CoV-2 strain which was used to design the vaccines currently in use. Thus, the protective capacity against the new variants was hard to predict.

Our observation is highly important since hemodialysis patients show reduced immunological responses to vaccination compared to healthy controls, which may be explained in the context of uremia [5,13]. The hemodialysis patients in our study showed a significantly increased capacity to neutralize both SARS-CoV-2 VoCs, Delta and Omicron BA.1, after the fourth vaccination. Our results are consistent with previous reports of significantly increasing anti-spike antibody titers after the fourth vaccination in hemodialysis patients [14,15] but add an important quality as these antibody titers were determined against the original vaccine strain of SARS-CoV-2 but not against the currently circulating variants. Furthermore, in line with previous work with a pseudovirus assay, we found a reduced neutralization capacity for VoC Omicron BA.1 compared to Delta [16].

Patients with a breakthrough infection between the first and the second blood sampling had a lower neutralization capacity for Omicron BA.1, only slightly missing significance. This was not seen for the Delta neutralization capacity. This might be partly explained by the fact that the analysis was performed between February and March 2022, when the Omicron wave peaked in Germany. Hence, over 99.3% of the majority of COVID-19 cases were Omicron infections at that time [10]. Logistic regression could not predict a SARS-CoV-2 breakthrough infection, possibly due to the low infection rate after the first blood collection. In French hemodialysis patients, a response towards wild-type virus neutralization two weeks after the third vaccination was present in approximately 54% of patients [5]. Another study in a British cohort found response rates of 97% and 72% for Delta and Omicron, respectively, in hemodialysis patients one month after the third BNT162b2 vaccination when applying an IC50 cut-off at 40 [17]. We found response rates of 57% for Omicron BA.1 and 79% for Delta four months after the third vaccination. Methodological differences in the neutralization assays [5,17] as well as time interval differences associated with reduced immune responses to vaccination [18] might explain these variations.

In line with previous reports [18,19,20], we identified immunosuppressive agents as a predictor for lower neutralization capacity, primarily prescribed to patients with a history of kidney transplantation. Patients on immunosuppressive medication had significantly lower neutralization capacity for Delta and, by trend, for Omicron BA.1. Other studies, however, did not identify immunosuppressive drugs as a predictor of neutralization capacity in hemodialysis patients [5]. Discrepancies might be explained due to the specific immunosuppressive agents prescribed. A previous study showed significantly reduced seroconversion rates in patients on anti-CD20 therapy regimes or mycophenolate mofetil, especially in combination with glucocorticoids [20], substances also prescribed to our patients.

Interestingly, a positive hepatitis B vaccination response was by trend associated with an improved neutralization capacity. This was, however, only seen for the Delta VoC. It thus needs to be determined by further studies if hepatitis B vaccination response might serve as a surrogate for COVID-19 vaccination response or vice versa.

In clinical routine, only NAb or anti-S antibody levels are readily and widely available. These, however, only detect the response against the original SARS-CoV-2 strains and not against the VoCs. Before the fourth vaccination, NAb was present in 96% of the study population, and response rates did not further increase after the fourth vaccination. However, when looking at the absolute change of NAb titers, NAb increased significantly after the fourth vaccination. This increase was less pronounced than the increase in IC50 values in infection neutralization due to the limited range of the assay, although the SARS-CoV-2 strain used for vaccination and in the NAb assay were identical. Although NAb levels are highly predictive of immune protection [21], this further demonstrates the limitation of routinely available assays.

We do not have outcome data of our cohort after the fourth vaccination concerning infection prevention but decreased COVID-19 incidence and severity in vaccinated hemodialysis patients have been observed by others [5]. Thus, increasing NAb levels might still be a good indicator of vaccine response after the fourth vaccination and, therefore, useful in clinical routine.

In a study by Espi et al., a third vaccination did not improve the immune response in patients that had already shown a high response after the second vaccination and was associated with more side effects [5]. In our cohort, we did not record side effects. Still, we observed even in NAb high-responder a further significant increase of neutralization capacity and, more importantly, a very strong increase in infection-neutralization capacity of the two most prevalent SARS-CoV-2 VoCs. Differences worth mentioning in the work of Espi et al. might be the application of a third dose three months after the second dose. At the same time, the fourth vaccination was administered at least four months after the third dose in our cohort. Nevertheless, reports of increased side effects in high-responders may argue for an individual decision-making process depending on routinely available antibody levels.

Finally, some limitations have to be mentioned. We examined the neutralization capacity of the Omicron sub lineage BA.1. The question remains if these results are generalizable to other Omicron subvariants currently becoming predominant. Further studies have to show if improved neutralization capacity after the fourth vaccination is associated with COVID-19 incidence and severity.

## 5. Conclusions

In conclusion, a fourth vaccination against SARS-CoV-2 significantly improves the antibody-mediated immune response in hemodialysis patients. A routinely applied four-time vaccination regimen, therefore, seems reasonable in hemodialysis patients. NAbs might be a good clinical surrogate of vaccination response. However, neutralization antibody titers above the upper limit of quantification should not hinder a fourth vaccination as this further improves and broadens live-virus infection neutralization.

## Figures and Tables

**Figure 1 vaccines-10-01328-f001:**
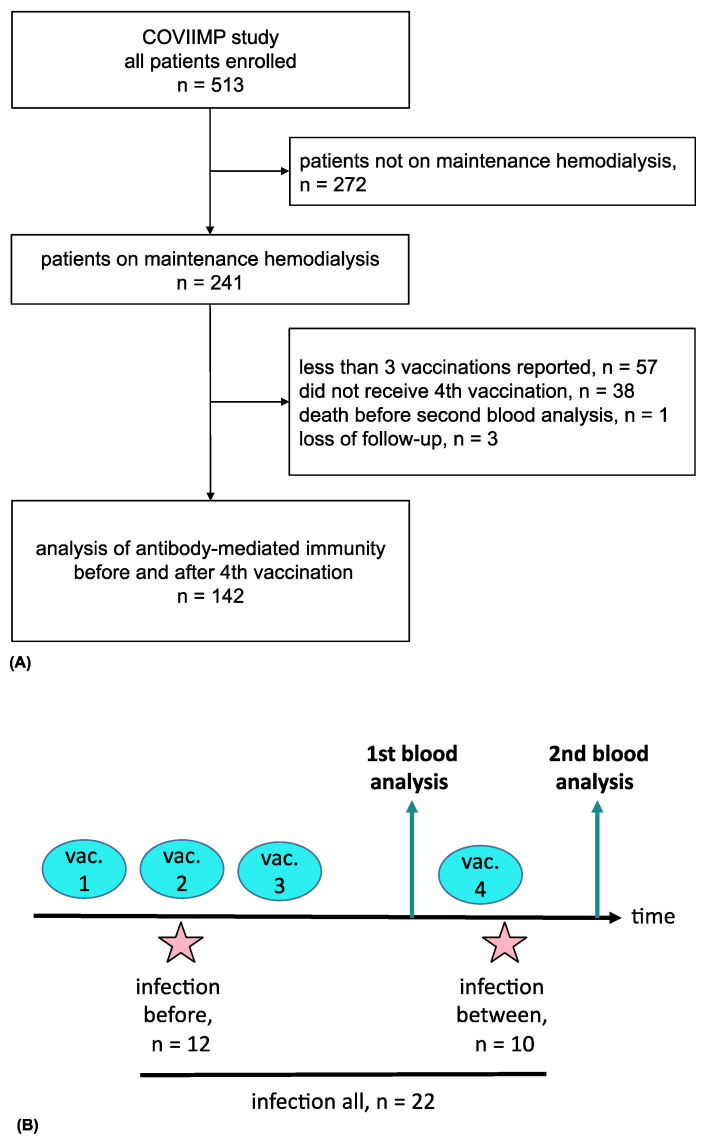
Flow chart of the COVIIMP study (**A**). Study design and observed SARS-CoV-2 infection cases (**B**). Abbreviations: vac., vaccination.

**Figure 2 vaccines-10-01328-f002:**
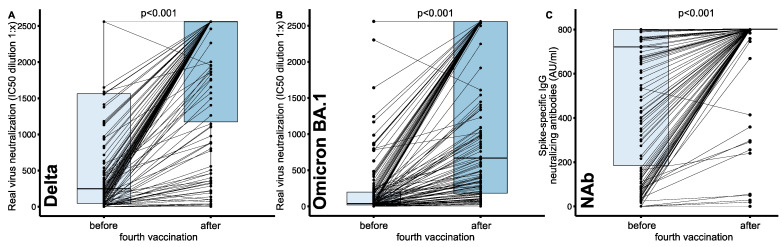
Changes in SARS-CoV-2 infection neutralization capacity before and after the fourth COVID-19 vaccination in hemodialysis patients. Real virus neutralization assay was performed using (**A**) the SARS-CoV-2 Delta (B.1.617.2) and (**B**) the Omicron (B.1.1.529, sub-lineage BA.1) variant of concern upon serial dilution of hemodialysis patient sera before and after the fourth vaccination. Inhibitory concentration (IC50) dilution values are given. (**C**) Change of spike-specific IgG neutralizing antibody (NAb) titers given in AU/mL in a surrogate neutralization assay. Dots indicate the measurement of an individual patient with lines connecting individual patient values before and after the fourth vaccination. Boxes indicate median and interquartile ranges. Statistical analysis was performed using paired-samples Wilcoxon test, *p* values indicate statistical significance between groups.

**Figure 3 vaccines-10-01328-f003:**
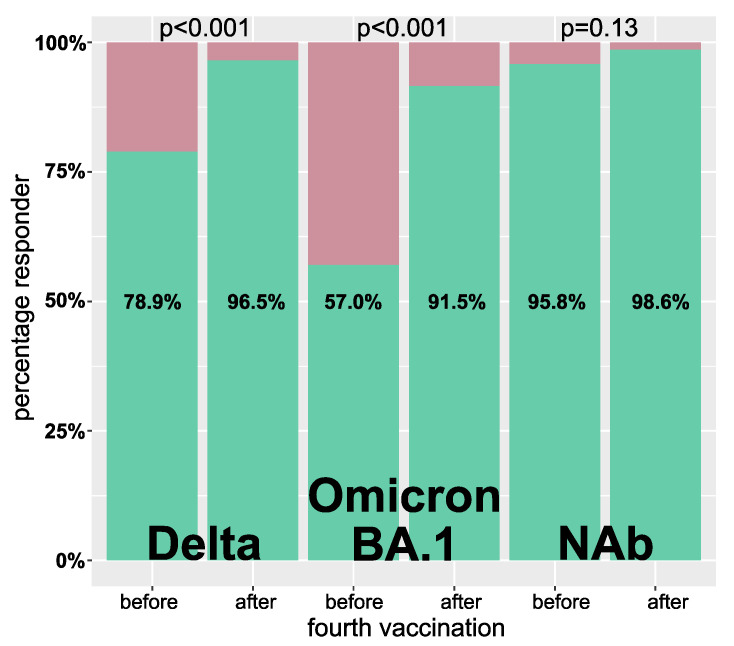
Percentage of responders before and after the fourth vaccination. A responder was defined by a Delta or Omicron BA.1 IC50 virus infection neutralization of >1:20 as well as neutralizing antibodies (NAb) ≥10 AU/mL. Green and red indicate the percentages classified as responder and non-responder, respectively. Statistical analysis was done using the McNemar test for paired samples.

**Figure 4 vaccines-10-01328-f004:**
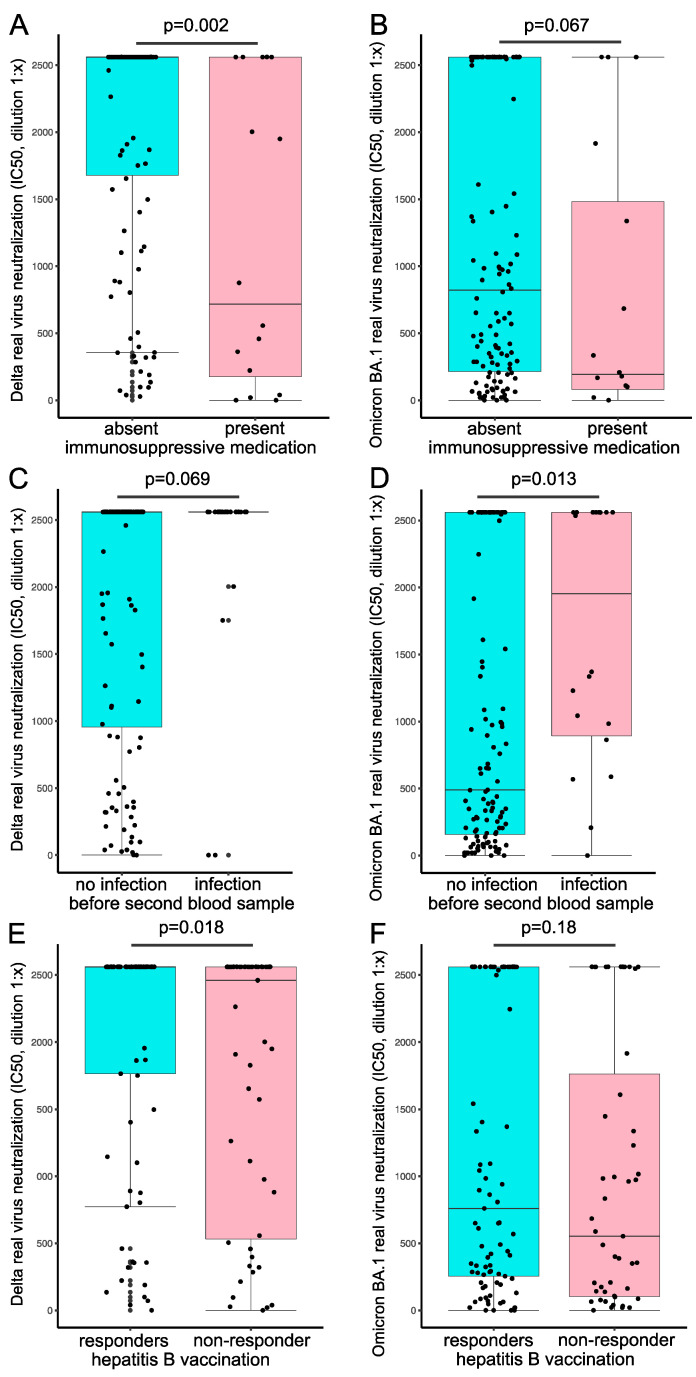
Influence of immunosuppressive medication, SARS-CoV-2 breakthrough infection, and hepatitis B response status on COVID-19 vaccine responses. Serum real-virus neutralization capacity for Delta (left column) and Omicron BA.1 (right column) was analyzed after the fourth vaccination in subgroups. Comparison of immunosuppressive drug treatment (**A**,**B**), the prevalence of SARS-CoV-2 infection before the second blood sampling (**C**,**D**), and hepatitis B vaccination non-response (**E**,**F**) on serum neutralization capacity. Statistical analysis was performed using the Mann-Whitney-U test, *p* values indicate statistical significance between groups.

**Figure 5 vaccines-10-01328-f005:**
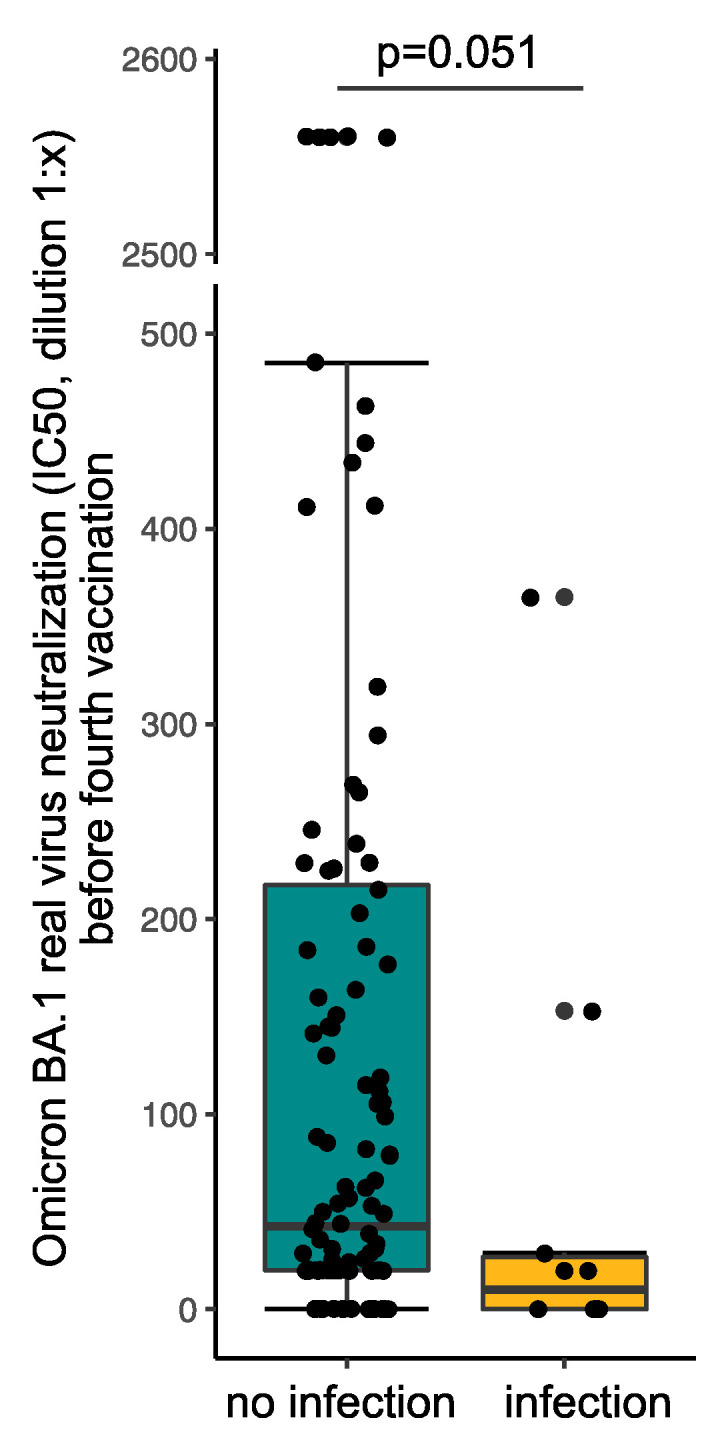
Serum neutralization capacity for Omicron BA.1 variant of concern before the fourth vaccination stratified by patients with SARS-CoV-2 breakthrough infections between the first and the second blood sampling. Statistical analysis was performed using the Mann-Whitney-U test, *p* value indicates statistical significance between groups. The y-axis is interrupted between 500 and 2500 for better visibility.

**Table 1 vaccines-10-01328-t001:** Patient characteristics.

		Omicron BA.1 Neutralization after Fourth Vaccination		
	Total *n* = 142	Low/Non-Responder *n* = 12	Responder *n* = 130	*p*
Age (years)	72.6 (61.5–80.6)	77.1 (67.0–79.7)	72.2 (60.5–80.6)	0.47
Female	48 (33.8%)	7 (58.3%)	41 (31.5%)	0.11
Dialysis vintage (months)	48.9 (21.3–83.7)	38.7 (13.4–63.6)	49.3 (21.9–84.0)	0.34
Vaccines				1.0
mRNA and vector	8 (5.6%)	0 (0.0%)	8 (6.2%)	
only mRNA	134 (94.4%)	12 (100.0%)	122 (93.8%)	
COVID-19 infection before second blood examination	22 (15.5%)	2 (16.7%)	20 (15.4%)	1.0
Time lap between infection and second blood examination (days)	215.7 ± 223.3	157.5 ± 222.7	224.6 ± 231.1	0.71
Charlson Comorbidity Index	5.0 (4.0–7.0)	5.5 (4.0–6.2)	5.0 (4.0–7.0)	0.95
History of kidney transplantation	16 (11.3%)	1 (8.1%)	15 (11.5%)	1.00
Immunosuppressive medication	16 (11.3%)	4 (33.3%)	12 (9.2%)	0.031
Hepatitis B vaccination non-response	51 (36.4%)	5 (41.7%)	46 (35.4%)	0.94
Renal diagnosis				
Glomerulopathy	22 (16.1%)			
Diabetic nephropathy	24 (17.5%)			
Hypertensive nephropathy	17 (12.4%)			
Congenital or cystic renal disease	13 (9.5%)			
Tubulointerstitial disease	2 (1.5%)			
Reflux nephropathy	3 (2.2%)			
Other	18 (13.1%)			
Nephropathy of unknown origin	43 (30.3%)			

Results are presented as mean (±SD) and median (interquartile range) for normally and non-normally distributed data, respectively; categorical data as total number (percentage). *p* values present the results of group-wise comparisons of patients neutralizing Omicron BA.1 after the fourth vaccination.

**Table 2 vaccines-10-01328-t002:** Univariate and multivariate regression models to identify predictors of Delta (A) and Omicron BA.1 (B) neutralization capacity, respectively as well as neutralizing antibodies (C) after the fourth vaccination.

	Univariate		Multivariate	
Predictor	*b* (95% CI)	*p*	*b* (95% CI)	*p*
**A.** **Delta**				
*(Intercept)*	-	-	1918.2 (985.5, 2850.9)	<0.001
Age (1 year)	−2.1 (−14.1, 10.0)	0.74	2.6 (−14.5, 19.6)	0.77
Dialysis vintage (1 month)	2.1 (−0.4, 4.6)	0.10	0.05 (−0.04, 0.13)	0.27
Charlson comorbidity index	−28.4 (−102.6, 45.7)	0.45	−17.3 (−120.8, 86.1)	0.74
Immunosuppressive medication	−814.7 (−1293.8, −355.9)	0.001	−867.3 (−1356.7, −377.9)	<0.001
Hepatitis B vaccination non-response	−331.9 (−658.1, −5.6)	0.046	−290.8 (−605.3, 23.7)	0.070
**B.** **Omicron BA.1**				
*(Intercept)*	-	-	1167.7 (91.6, 2243.8)	0.034
Age (1 year)	−0.5 (−13.9, 12.9)	0.94	0.2 (−19.4, 19.9)	0.98
Dialysis vintage (1 month)	1.0 (−1.9, 3.8)	0.50	0.02 (−0.07, 0.12)	0.62
Charlson comorbidity index	−7.9 (−90.5, 74.8)	0.85	−0.6 (119.9, 118.7)	0.99
Immunosuppressive medication	−382.7 (−933.3, 167.9)	0.17	−457.6 (−1031.3, 116.0)	0.12
Hepatitis B vaccination non-response	−228.1 (−590.7, 134.4)	0.22	−180.7 (−568.3, 206.8)	0.36
**C.** **Neutralizing antibodies**				
*(Intercept)*	-	-	837.9 (661.4, 1014.4)	<0.001
Age (1 year)	−1.2 (−3.6, 1.1)	0.30	−1.3 (−4.6, 1.9)	0.41
Dialysis vintage (1 month)	0.4 (−0.1, 0.19)	0.12	0.01 (−0.01, 0.02)	0.32
Charlson comorbidity index	−7.0 (−21.3, 7.4)	0.34	2.8 (−16.7, 22.3)	0.78
Immunosuppressive medication	−209.6 (−302.1, −117.0)	<0.001	−223.0 (319.9, −126.0)	<0.001
Hepatitis B vaccination non-response	−228.1 (−590.7, 134.4)	0.22	−22.8 (−86.5, 40.9)	0.48

Abbreviations: *b*, regression coefficient; CI, confidence interval.

## Data Availability

The datasets for this manuscript are not publicly available because written informed consent did not include wording on data sharing (German data protection laws). Reasonable requests to access the datasets should be directed to the corresponding author.

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
