# Peer review of "Improved SARS-CoV-2 Neutralization of Delta and Omicron BA.1 Variants of Concern after Fourth Vaccination in Hemodialysis Patients"

_vaccines, 2022, doi:10.3390/vaccines10081328_

Round 1

Reviewer 1 Report

This manuscript clearly shows that hemodialysis patients have a pronounced advantage from a fourth vaccination against SARS-CoV-2. This study reports on 142 hemodialysis patients until March 2022. The immune response was measured by a surragate neutralization test as well as by deta and omicron-specific direct neutralization assays. The neutralization titers were markedly increased after the fourth vaccination. The surrogte test showed exaggerated results; the direct culture-based neutralization tests provide much more relevant results.

The study is well explained and very well presented. There is no relevant need for modification of the manuscript. The results are highly relevant for dialysis patients and their physicians.

Author Response

We thank reviewer 1 for the positive feedback.

Reviewer 2 Report

Regarding SARS-CoV-2, the authors reported the plasma anti-delta and anti-omicron SA.1 neutralization antibody (NAb) titers and by a surrogate ELISA, plasma NAb titers to the Wuhan strain, in N=142 hemodialysis patients before and after a 4th COVID-19 vaccination.  The major findings are: 1) Neutralizing titers were significantly increased against the Wuhan strain, delta, and omicron SA.1 after a 4th vaccination than before. 2) Immunosuppressant was associated with poorer COVID-19 vaccine responses. 3) Hepatitis B non-responders were associated with poorer COVID-19 vaccine responses.  4) Breakthrough infections were associated with poorer responses before and after the 4th vaccination. Based on these findings, the authors reached the recommendation of a 4th COVID-19 vaccination for hemodialysis patients.  In general, the data supported the conclusions.  Because BA.1 is not representative of the antibody invasion by Omicron (PMID: 35016198; PMID: 35240676; PMID: 35790190), it would be helpful to specify Omicron BA.1 (not just Omicron) throughout the manuscript.  Also, it should be made clear at the beginning that the comparison was not about the 3rd and 4th vaccinations, but ~4 months after the 3rd vaccination (responses waning) and ~4 weeks after the 4th vaccination (responses peaking).  Nonetheless, it is useful to report that the 4th vaccination is able to boost the waning responses in the majority of hemodialysis patients.

Author Response

We thank the Reviewer for the comments and agree with the Reviewer. We specified the variant of concern Omicron BA.1 throughout the manuscript including the title, when appropriate. All changes made are highlighted by track-change.

Furthermore, we improved one sentence in the discussion section:

„The strength of our study is the examination of live-virus infection neutralization capacity of patients’ sera for two of the most recent SARS-CoV-2 VoCs, Delta and Omicron BA.1.” (page 11, line 341 ff)

Additionally, we added the sublineage BA.1 to the Figures 2, 3, 4, and 5 to allow a clear understanding of the variants examined on first sight.

Concerning the second point, we added specifications to the abstract as well as the introduction in order to clarify the timing of the analyses right from the beginning, namely four months after the third vaccination compared to four weeks after the fourth vaccination.

Abstract: „Four weeks after the fourth vaccination serum infection neutralization capacity significantly increased from a 50% inhibitory concentration (IC50, serum dilution factor 1:x) of 247.0 (46.3-1560.8) to 2560.0 (1174.0-2560.0) for the Delta VoC, and from 37.5 (20.0-198.8) to 668.5 (182.2-2560.0) for the Omicron VoC (each p<0.001) compared to four months after the third vaccination.“ (page 2, line 43 ff)

Introduction: „Here, we present the results of the live-virus infection neutralization of SARS-CoV-2 Delta and Omicron BA.1 VoCs and antibody-mediated immunity shortly before compared to four weeks after the fourth COVID-19 vaccination in a cohort of 142 hemodialysis patients.“ (page 3, Line 83 ff)

Reviewer 3 Report

This is an excellently presented original article with the introduction explaining clearly on the current climate of concerns from the pandemic. Considering the emergence of delta and omicron viruses, and that there is a global movement towards achieving four COVID-19 vaccinations across populations to counter this amongst other variants, there is a basis for the authors to conduct this work. This work is valuable in the context of the hemodialysis cohort who are a vulnerable group with poorer outcomes with COVID-19 infection. 

The authors have presented the methodology clearly with selection of study sample, apparatus for and methods of data collection and reasoning of parameters used to gather results explained explicitly. Results were well supported with appropriate tables and figures to guide clearer interpretation. The discussion section was flowing throughout, with suitable references made towards important research works conducted in this topic for the first three vaccination cycles. Throughout the manuscript, I liked that the authors made reference towards hepatitis B vaccination response and its role as a potential surrogate for COVID-19 vaccination response and vice-versa. 

Limitations of the work were mentioned suitably, and the conclusion was clear in advocating for a fourth vaccination improving antibody-mediated immune response in the hemodialysis cohort. 

No issues with English Language throughout

Author Response

We thank reviewer 3 for the positive feedback.